# Effect of Oat β-Glucan on Affective and Physical Feeling States in Healthy Adults: Evidence for Reduced Headache, Fatigue, Anxiety and Limb/Joint Pains

**DOI:** 10.3390/nu13051534

**Published:** 2021-05-01

**Authors:** Thomas M. S. Wolever, Maike Rahn, El Hadji Dioum, Alexandra L. Jenkins, Adish Ezatagha, Janice E. Campbell, YiFang Chu

**Affiliations:** 1Formerly GI Labs, INQUIS Clinical Research, Ltd., Toronto, ON M5C 2N8, Canada; alexandrajenkins@inquis.com (A.L.J.); aezatagha@inquis.com (A.E.); jcampbell@inquis.com (J.E.C.); 2Quaker Oats Center of Excellence, PepsiCo R&D Nutrition, Barrington, IL 60010, USA; mrinny96@gmail.com (M.R.); ElHadji.Dioum@pepsico.com (E.H.D.); yifang.chu@pepsico.com (Y.C.)

**Keywords:** randomized clinical trial, humans, symptoms, gastrointestinal tract, musculo-skeletal system, oats, oatmeal, dietary fiber, beta-glucan

## Abstract

The gastrointestinal (GI) side-effects of dietary fibers are recognized, but less is known about their effects on non-GI symptoms. We assessed non-GI symptoms in a trial of the LDL-cholesterol lowering effect of oat β-glucan (OBG). Participants (*n* = 207) with borderline high LDL-cholesterol were randomized to an OBG (1 g OBG, *n* = 104, *n* = 96 analyzed) or Control (*n* = 103, *n* = 95 analyzed) beverage 3-times daily for 4 weeks. At screening, baseline, 2 weeks and 4 weeks participants rated the severity of 16 non-GI symptoms as none, mild, moderate or severe. The occurrence and severity (more or less severe than pre-treatment) were compared using chi-squared and Fisher’s exact test, respectively. During OBG treatment, the occurrence of exhaustion and fatigue decreased versus baseline (*p* < 0.05). The severity of headache (2 weeks, *p* = 0.032), anxiety (2 weeks *p* = 0.059) and feeling cold (4 weeks, *p* = 0.040) were less on OBG than Control. The severity of fatigue and hot flashes at 4 weeks, limb/joint pain at 2 weeks and difficulty concentrating at both times decreased on OBG versus baseline. High serum c-reactive-protein and changes in c-reactive-protein, oxidized-LDL, and GI-symptom severity were associated with the occurrence and severity of several non-GI symptoms. These data provide preliminary, hypothesis-generating evidence that OBG may reduce several non-GI symptoms in healthy adults.

## 1. Introduction

To protect against malnutrition and non-communicable diseases global nutrition recommendations include advice to consume more whole grains to help maintain an adequate intake of dietary fiber [1]. Oats are a whole grain containing the soluble dietary fiber β-glucan [2]. Oat β-glucan (OBG) reduces serum LDL-cholesterol [3,4] and postprandial glycemic responses [5], physiological effects associated with reduced risk for cardiovascular disease [6] and type 2 diabetes [7], respectively. Dietary fibers such as OBG are not absorbed in the small intestine and reach the colon where they increase the mass of the colonic contents and may be partly or completely fermented by colonic microbiota, effects which have desirable and undesirable consequences as explained below.

The undesirable side-effects of dietary fibers include abdominal bloating and pain, flatus and diarrhea which, although transitory in nature, may limit the acceptability of high fiber foods [8]. Consistent with this were our findings in healthy subjects who consumed 3 g/day OBG or Control for 4 weeks. After 2 weeks, flatulence and abdominal discomfort had increased from baseline on both the OBG and Control treatments; but by 4 weeks these symptoms had begun to return to baseline [9]. In a cross-sectional study of adults with coeliac disease, higher long-term oat intake correlated with lower gastrointestinal symptom scores [10].

The desirable effects of increased fiber intake include increased stool bulk which may be beneficial for conditions such as constipation, irritable bowel syndrome [11] and diverticulosis [12] although not all types of fiber are equally effective. Oats and oat bran appear to increase stool weight per gram of dietary fiber to a similar extent as other sources of fiber [13]. Oats also decrease fecal pH (a marker of increased short-chain fatty acid (SCFA) production) and increase the growth of beneficial gut microbiota [14]. Plausible mechanisms have been proposed whereby the stimulation of SCFA production and alteration of the colonic microbiome can influence systemic oxidative stress and inflammation and thereby influence both physical and mental functioning [15,16,17]. In this context, it is of interest that higher self-reported fiber intake was associated with higher positive affect in healthy children [18] and, in a randomized, cross-over study, consumption of a high fiber breakfast cereal for 2 weeks was associated with less fatigue compared to control in healthy adults [19]. We found that participants with type 2 diabetes treated by diet alone who consumed a low glycemic index diet containing 37 g/day fiber for 1 year experienced less severe headaches, less severe pains in joints or limbs and less severe gloomy thoughts than those on the lower fiber (21–23 g/day) diets [20]. OBG has been shown to enhance the endurance capacity of rats and influence serum metabolites in such a way as to suggest anti-fatigue properties [21], but there is a paucity of information about the effect of OBG on non-gastrointestinal symptoms in humans. To address this, we assessed a panel of non-gastrointestinal symptoms as tertiary objectives in a study the primary purpose of which was to determine if a novel oat product containing OBG would reduce serum LDL-cholesterol [9].

## 2. Materials and Methods

We performed a randomized, double-blind, placebo controlled, parallel arm design clinical trial at INQUIS Clinical Research, a contract research organization. The protocol was approved by the Western Institutional Review Board^®^ (Puyallup, WA, USA) and all participants provided informed consent by signing the IRB approved consent form. The study was registered at www.clinicaltrials.gov (accessed on 11 April, 2019) with identifier: NCT03911427.

Participants were healthy males and non-pregnant, non-lactating females without diabetes, aged 18–65 year, with fasting calculated LDL-cholesterol between 3.00 and 5.00 mmol/L, inclusive, and BMI between 18.5 to <40 kg/m^2^. Details of the inclusion/exclusion criteria can be found in Appendix A.

After signing the consent form, participants were screened over a period of 2–4 weeks involving 2 visits to the clinic. First they answered questions about their medical and drug history and had their height, weight and blood pressure measured, then they gave a fasting blood sample and were given instructions about how to record a 3-day diet record (3DDR) [22,23,24]. Participants who were eligible based on the results of the screening blood sample were contacted and asked to start recording a 3DDR and attend at the clinic to review the 3DDR and fill out the symptoms questionnaire (second screening visit, Screen 2). To avoid the potential confounding effects of high intakes of saturated fat or dietary fiber on the results, participants were excluded if their intake of saturated fat was ≥15% of energy or they were consuming >14 g/1000 kcal dietary fiber. Eligible participants were contacted by telephone or e-mail, advised to follow their usual dietary and exercise habits and to refrain from consuming oat, barley and psyllium products for the duration of the study and an appointment was made for the baseline visit.

### 2.1. Study Visits

Eligible participants visited the clinic after 10–14 h overnight fasts at baseline (week 0) and at 2 and 4 weeks, having been asked to avoid alcohol and unusual levels of physical activity and food intake for 24 h before each visit. At each visit medications and adverse events were reviewed, weight and blood pressure were measured, a fasting blood sample obtained and a symptoms questionnaire filled out. At baseline, participants were randomly assigned to either the Test or Control treatment. After the blood sample at each visit participants consumed the first daily treatment sachet and were given a snack or light breakfast. At baseline and week 2 participants were provided with a sufficient supply of treatment sachets to last until the next visit and a shaker cup and whisk with which to mix their assigned treatment and were given a study diary in which to record sachet consumption. Compliance was checked at week 2 and week 4 by counting unused sachets and, at week 2, participants were provided with forms and instructions to record another 3DDR before the next visit. Symptom questionnaires were filled out during each of the visits at Screen 2, Baseline, 2 weeks and 4 weeks.

### 2.2. Interventions

Interventions were provided in color-coded sachets: the Test intervention consisted of an oat ingredient delivering 1 g of β-glucan sachet; the Control intervention was a rice milk powder containing 0 g β-glucan. Each sachet contained 0.9 g fat (0 g saturated), 1.9 g protein, ~14 g available carbohydrate and 1.9 g (Test) or 0.3 g (Control) dietary fiber. Detailed nutrition information is given in Appendix A. Each sachet was mixed with 8 oz (240 mL) of cold water, shaken for 30 sec and consumed immediately. The Test and Control beverages had similar look, taste and smell. Subjects were asked to consume 3 sachets daily on 3 separate occasions separated by at least 3 h and preferably immediately before or within 10 min of each main meal (breakfast, lunch and dinner).

### 2.3. Symptoms Questionnaire

Participants were asked to rate the severity of each of 27 symptoms (11 gastrointestinal (GI) symptoms and 16 non-GI symptoms) they experienced over the previous 2 weeks on a 4-point scale: none = 0, mild = 1, moderate = 2 or severe = 3. We used this questionnaire in 2 previous studies in patients with diabetes [20,22,23]. The results for all 16 GI symptoms have been reported elsewhere [9] but the sum of the scores for the 5 most prevalent (major) GI symptoms (flatulence, diarrhea, constipation, abdominal distention and abdominal pain) are included below. Of the 16 non-GI symptoms the following 4 were experienced by a maximum of less than 10% of participants at any visit (baseline, 2 weeks and 4 weeks, respectively) and the results are not presented here: Increased appetite (6%, 9% and 5%), Palpitation/Throbbing of heart (4%, 6%, 4%), Balance disturbances (5%, 3%, 5%) and Numbness/burning/itching hands feet (8%, 4%, 5%). The remaining 12 symptoms were as follows: Headache, Fatigue, Lack of appetite, Tend to become exhausted (Exhaustion), Feelings of anxiety, Lack of energy, Pains in joints or limbs (Limb/joint pain), Diminished ability to concentrate (Reduced ability to concentrate), Feeling cold, Hot flashes/sensation of rising heat (Hot flashes), Gloomy thoughts and Inner tension.

### 2.4. Data Analysis, Management and Calculations

Blood samples for blood lipids and lipoproteins, glucose and insulin were collected at Baseline, 2 weeks and 4 weeks, and for glycated albumin, high sensitivity c-reactive protein (CRP) and oxidized LDL (oxLDL) at Baseline and 4 weeks and the results presented elsewhere [9]. However, results for CRP (analyzed by LifeLabs, Inc., Mississauga, ON, USA) and oxLDL (analyzed by ELISA kit, catalog #30-7810, Alpco Diagnostics, Salem, NH, USA) are included here because of their potential association with affective and physical symptoms. Each 3DDR was reviewed with the participant by a dietitian or nutritionist and subsequently analyzed for nutrient content (ESHA Food Processor, ESHA Research, Salem, OR, USA). The nutrients contained in the treatment sachets were added to those from foods recorded on the 3DDR based on the sachet consumption counts

### 2.5. Power Analysis

The sample size calculation assumed that the SD of the change in LDL-cholesterol between baseline and 4 weeks would be 0.455 mmol/L [24]. Using this SD, *n* = 90 subjects per arm provides 80% power to detect an 0.19 mmol/L difference (5%) in LDL-cholesterol change between the test and control arms. To improve the likelihood that at least *n* = 180 would complete the study we allowed for to 15% dropouts and enrolled 207 subjects.

### 2.6. Statistical Analysis

Included in this analysis are all 191 subjects who were randomized and completed the study with no protocol violations and with 90% compliance (76/84 sachets over 4 weeks) based on the sachet count. The highly skewed distributions of CRP and oxLDL values were normalized by log transformation. Categorical measures were summarized as frequencies. To determine whether the prevalence of symptoms during treatment differed from that at the Baseline visit, and differed between treatments, the proportions of subjects who experienced any symptom (mild, moderate or severe) were compared using the chi-square test. To determine if the severity of symptoms during treatment differed from those before treatment, the ratio of the number of subjects in each treatment group whose scores at 2 weeks and 4 weeks were greater (more severe, M) or less (less severe, L) than the median score before treatment (defined as the median of the scores at the second screening (Scr2) and Baseline visits) were compared to an expected ratio of 1:1 by chi-square test. To determine if the change in severity of symptoms differed between treatments, the ratios of L:M on Test and Control were compared by Fisher’s exact test [20]. Similarly, the severity of symptoms in participants whose serum CRP, serum oxLDL and the sum of the scores for 5 major GI symptoms (flatulence, diarrhea, constipation, abdominal bloating, abdominal discomfort) increased versus those in whom they did not increase during treatment.

## 3. Results

Recruitment began in April 2019 and the last subject visit occurred in February, 2020. The recruitment questionnaire was completed by 2607 individuals of whom 1690 were invited for screening; 538 attended Screen 1 (243 ineligible, 15 lost and 16 withdrawn), 264 attended for Screen 2 (7 excluded for SFA intake > 15% en, 10 excluded for dietary fiber intake >14 g/1000 kcal, 8 lost and 15 withdrawn) and 224 were randomized (17 withdrawn, 103 received Control and 104 received Test). Four (4) Control participants were withdrawn early for non-compliance (*n* = 3) or a serious adverse event (not related to treatment, *n* = 1) and 4 participants were withdrawn after completion for non-compliance (*n* = 2) or antibiotic use (*n* = 2) leaving *n* = 95 Control participants who completed the study per protocol. Similarly, 3 Test participants were withdrawn early for antibiotic use (*n* = 1) or scheduling difficulties (*n* = 2) and 5 were withdrawn after completion for non-compliance (*n* = 2), antibiotic use (*n* = 2) or insulin use (*n* = 1) leaving *n* = 96 Test participants who completed the study per protocol. The study flow chart and further details are given elsewhere [9].

The per-protocol population consisted of 72 males and 119 females of whom 102 were Caucasian, 39 South Asian, 14 African, 14 Hispanic, 8 East Asian, 6 South-east Asian, 4 West Asian, 3 mixed and 1 Indigenous; participant sex and ethnicity did not differ significantly on Test vs. Control. Participants were aged (mean ± SD) 47.6 ± 11.4 years, BMI 27.9 ± 4.6 kg/m^2^ (BMI 25.0–<30, *n* = 191; BMI 30.0–<35, *n* = 132; BMI 35.0–<40, *n* = 43), total cholesterol 5.83 ± 0.70 mmol/L, triglycerides 1.46 ± 0.69 mmol/L, HDL-cholesterol 1.44 ± 0.41, LDL-cholesterol 3.73 ± 0.50, fasting glucose 4.95 ± 0.51 with no significant differences between Test and Control

Mean energy intake increased from baseline on Test and Control by approximately the amount of energy the sachets contained (Figure 1). Sugars intakes, increased to an equivalent extent on both treatments. Starch intake did not change significantly on Test but increased from baseline on Control. Dietary fiber intake increased during treatment on Test by approximately the amount in the Test sachets and the difference was significant compared both to baseline and to Control (Figure 1). Total fat and protein intakes increased from baseline on Control, but not Test; alcohol intake decreased from baseline on Test.

### 3.1. Gastrointestinal (GI) Symptoms

The occurrence and severity of GI symptoms are presented in detail elsewhere [9]. One or more major GI symptom (flatulence, diarrhea, constipation, abdominal distention, and abdominal pain) was present in 36% of Test and 28% of Control participants at baseline (ns); the prevalence increased to 61% and 52%, respectively, at week 2 and fell to 47% and 42% by week 4 (differences between Test and Control were not significant). Furthermore, the sum of the scores for the 5 major GI symptoms did not differ between Test and Control, respectively, at any visit with mean ± SD as follows: screen 2, 0.95 ± 1.48 vs. 0.81 ± 1.55; baseline, 0.77 ± 1.39 vs. 0.53 ± 1.29; week 2, 1.67 ± 2.12 vs. 1.45 ± 2.09; week 4, 1.20 ± 1.83 vs. 0.91 ± 1.39.

### 3.2. Non-GI Symptoms

The number of subjects reporting symptom severity of none, mild, moderate or severe for each visit on Test and Control are shown in Appendix A. The prevalence and severity of the 12 most common symptoms are shown in Figure 2.

The presence or severity of 7 of the symptoms at Baseline did not differ significantly by age, sex or BMI. The presence of Limb/Joint pain at Baseline differed by sex (females > males) and age (over 48 years > under 48 years) and the severity of Limb/Joint pain was greater in obese compared to lean or overweight subjects. The presence of fatigue and hot flashes were affected by sex (females > males) and age (over 48 years > under 48 years). The severity of lack of energy was greater in females than males, whereas feeling cold was more common in males than females (Appendix A).

The prevalence of symptoms did not differ significantly between Test and Control for any symptom at any time. However, on Test the prevalence of fatigue was lower at week 4 and exhaustion at week 2 compared to baseline (Figure 2B,F). The change in severity of symptoms did not differ significantly between Test and Control except for headache and anxiety (less severe on Test vs. Control at week 2) and feeling cold at week 4 (insets on Figure 2A,D,L). Compared to before treatment (median of scores at the Screen 2 and Baseline visits), symptoms were less severe as follows: fatigue (Test at week 4), lack of energy (both Test and Control at week 2 and week 4), exhaustion (Test at week 2 and week 4 and Control at week 4), limb/joint pain (Test at week 2), hot flashes (Test at week 4), feeling cold (Control at week 2 and Test and week 4); anxiety and inner tension (Test and Control at week 2 and week 4); and reduced ability to concentrate (Test at week 2 and week 4) (Figure 2). If the severity of symptoms at during treatment is compared to that at the Baseline visit alone (Appendix A), fewer participants have more severe or less severe symptoms, but the direction of the effects are similar to those shown in Figure 2.

The presence of symptoms at baseline was associated with increased serum CRP (mg/L) (median [25%, 25%]) as follows (absence vs. presence, respectively): lack of appetite, 1.31 [0.66, 3.36], *n* = 167 vs. 2.52 [1.48, 4.15] *n* = 24, *p* = 0.013; exhaustion, 1.30 [0.65, 3.40], *n* = 138 vs. 2.12 [1.07, 3.91], *n* = 53, *p* = 0.005; and limb/joint pain, 1.30 [0.65, 3.03], *n* = 127 vs. 2.11 [1.08, 4.12], *n* = 64, *p* = 0.009. Although the presence of headache at baseline was not associated with higher serum CRP, in participants whose CRP did not increase during the study the severity of headache was greater on Control than Test (*p* = 0.049; Figure 3A). The severity of exhaustion was greater in participants whose CRP increased vs. those in whom it did not, however, changes in CRP did not affect the difference between Test and Control (Figure 3B).

Serum oxLDL was not associated with the presence of any symptom at baseline. The severity of fatigue was similar for participants in whom oxLDL increased during treatment compared to those in whom it did not. However, in participants whose oxLDL did not increase, fatigue was more severe on Control than Test (*p* = 0.052, Figure 3C). The severity of hot flashes was greater in participants in whom oxLDL increased compared to those in whom it did not (*p* = 0.049), but the severity of hot flashes on Test was similar to Control regardless of the change in oxLDL (Figure 4D).

At baseline, 11 of the 12 non-GI symptoms were significantly more common at in the *n* = 88 participants with GI symptoms than in the *n* = 103 without GI symptoms (Table 1). Additionally, changes in the severity of GI symptoms during the trial were associated with differences in the severity of several non-GI symptoms. The overall severity of headache was not associated with the severity of GI symptoms at either 2 weeks or 4 weeks. However, at 2 weeks, headache was less severe on Test vs. Control in participants with no increase in the severity of GI symptoms (*p* = 0.021, Figure 4A). The overall severity of fatigue at 2 weeks was lower in participants whose GI symptoms became more severe compared to those in whom they did not (*p* = 0.027). The same was true for participants on the Control treatment (*p* = 0.017). However, the severity of Fatigue on Test treatment was not associated with differences in the severity of GI symptoms (Figure 4B). The overall severity of joint/limb pain was greater at 2 weeks in participants with more severe vs. the same or less severe GI symptoms (*p* = 0.046). The same was true for Control participants at 2 weeks (*p* = 0.011) and 4 weeks (*p* = 0.049) but not for Test participants (ns). At 2 weeks, the severity of joint/limb pain was lower on Test vs. Control in participants with more severe GI symptoms than at baseline (*p* = 0.038, Figure 4C).

## 4. Discussion

The present results provide hypothesis-generating evidence that the OBG-enriched oat product tested may influence several affective and physical feeling states in healthy adults with LDL cholesterol between 3 and 5 mmol/L. Furthermore, markers of systemic inflammation and oxidative stress and the severity of GI symptoms modified some of these effects. These findings are consistent with current concepts about how alterations in the gut-brain axis (increased colonic fermentation and alteration of the gut microbiota) influence metabolism, behavior and brain function by a variety of different mechanisms including effects on afferent pathways between the gut and the brain, short-chain fatty acid (SCFA) production and other microbial by-products and metabolites which have local and systemic effects on gut hormones, oxidative stress and inflammation [17,25].

The association between the presence of GI symptoms and the increased prevalence of 11 of the 12 non-GI symptoms is not a new finding and could be ascribed to psychological or physiological factors. For example, it is known that the perception of GI symptoms such as constipation and flatulence are not related to the amount of gas in the colon [26] and are associated with personality and anxiety [27]. Thus, participants who noticed GI symptoms may have been, in general, more aware of other abnormal sensations and feelings they experienced and were more able to remember to report them. However, the association between GI and non-GI symptoms could have a physiological basis via the gut-brain axis. There is abundant evidence from studies in vitro and in animal models that oats and OBG influence SCFA production and the colonic microbiome [14], although the results are inconsistent and there are few studies in humans. OBG-enriched oat bran containing 20 g/d dietary fiber and 10 g/d OBG did not increase fecal SCFA after 4 weeks but did so at 8 and 12 weeks in healthy subjects [28]. Furthermore, fecal SCFA were not increased by either granola containing 6 g/d fiber and 3 g/d OBG daily for 5 weeks [29] or by 60 g oatmeal containing 8.5 g dietary fiber and 4.7 g OBG daily for 1 week [30]. However, the granola increased fecal bifidobacteria and lactobacilli [29] and the oatmeal influenced bacterial metabolism as judged by reduce β-galactosidase and urease concentrations [30]. A recent study showed, in hypercholesterolemic subjects that compared to 80 g white rice, 80 g oatmeal daily for 45 days increased the abundance of fecal Firmicutes [31]. The effect of OBG on the fecal microbiome might also by its molecular weight, since 3 g high molecular weight (1349 kDa) barley β-glucan for 5 weeks significantly increased fecal Bacteriodetes and reduced Firmicutes compared to a β-glucan-free control and to 3 g low molecular weight (288 kDa) barley β-glucan [32]. Thus, there is evidence in humans that the amount of OBG we provided could influence the fecal microbiome, but not fecal SCFA. However, the composition of the fecal microbiome may not reflect that of the mucosa-associated micro-organisms [33]. In addition, there is evidence in humans that increased SCFA production can occur in the absence of a change in fecal SCFA [34] and that a high fecal acetate concentration may reflect reduced acetate absorption rather than increased production [35].

The severity of headache was lower on Test than Control at 2 weeks in all subjects (Figure 2A) and at 4 weeks in those in whom serum CRP did not increase (Figure 3A). The causes of headaches are not well understood, but may include hypoglycemia [36], obesity [37] or inflammation [25]. However, the *n* = 67 participants with headache at baseline had similar (mean ± SD) BMI, 28.08 ± 4.70 kg/m^2^, and median CRP, 1.52 mg/L, as the *n* = 127 without headache, 27.89 ± 4.58 kg/m^2^, and 1.53 mg/L.

Fatigue became less common and less severe than at baseline after 4 weeks treatment with OBG, as did exhaustion after 2 weeks, but the differences from control were not significant. Lack of energy was less severe on both Test and Control treatments compared to baseline. These results are in line with studies showing that OBG [21] and konjac oligosaccharide [38] increased maximum exercise time in rats or mice and altered metabolites suggestive of reduced fatigue. In humans, consuming cereals high in wheat-fiber for 2 weeks reduced fatigue compared to control in healthy adults [19] and 13 weeks treatment with a prebiotic containing inulin and fructo-oligosaccharides reduced exhaustion in elderly subjects when compared to control [39]. Fatigue is associated with lack of sleep, an association that may be mediated by increased inflammation [40]. At baseline, serum CRP was similar in participants with and without fatigue (not shown) and with or without lack of energy (not shown), but the presence of exhaustion at baseline was associated with a higher CRP. Furthermore, the severity of exhaustion increased from baseline to a greater extent in participants whose CRP increased compared to those in whom it did not (Figure 3B).

Pains in muscles can be due to overuse or to reduced circulation, while pain in joints is most commonly due to intervertebral disc herniation or osteoarthritis, both of which are associated with obesity, chronic inflammation [41,42,43] and older age. This is consistent with our data in that the *n* = 64 participants with limb/joint pain at baseline were older (mean ± SD, 52.1 ± 10.3 vs. 45.3 ± 11.3 years, *p* < 0.001), and had higher BMI (29.1 ± 4.5 vs. 27.3 ± 4.6 kg/m^2^, *p* = 0.013), and higher CRP (2.11 [1.08, 4.12] vs. 1.30 [0.65, 3.03] mg/L, *p* = 0.009) than the *n* = 127 without limb/joint pain. After 2 weeks treatment, limb/joint pain was less severe than at baseline in all subjects on OBG (Figure 2C), and, in participants with more severe GI symptoms, less severe on OGB compared to control at 2 weeks (Figure 4C). On Control, but not OBG, having more severe GI symptoms was associated with more severe limb/joint pain at 2 and 4 weeks (Figure 4C). These observations, along with the increased prevalence of limb/joint pain in those with GI symptoms (Table 1) suggest a link between the gut and joint/limb pain. There is some evidence that people with rheumatoid arthritis (RA) have different colonic microbiota and that probiotic treatment can improve the condition in the absence of changes in CRP [44] but this may not apply to our study population, most of whom likely did not have RA.

During the study, the severity of anxiety decreased from baseline on both treatments, with the reduction on OBG being greater than that on Control at 2 weeks (*p* = 0.059, Figure 2D). A link between the colonic microbiota and anxiety is suggested by a study in which a probiotic formulation containing *Lactobacillus helveticus* R0052 and *Bifidobacterium longum* R0175, given to male Wistar rats for 2 weeks, reduced anxiety-like behavior, and given to humans for 30 d reduced psychological distress, the Hospital Anxiety and Depression Scale and urinary free-cortisol excretion [45]. Additionally, compared to control, 5.5 g/d galactooligosaccharides for 3 weeks, reduced the salivary cortisol waking response and increased the processing of positive vs. negative attentional vigilance in healthy subjects, although these effects were not seen with fructooligosaccharides [46].

The prevalence of feeling cold at 4 weeks did not change from baseline on either treatment, but the severity of feeling cold at 4 weeks was less on Test than Control (Figure 2L). Consistent with several studies in which feeling cold was associated with musculoskeletal pain [47,48,49], we found that feeling cold was present in 41.9% of 43 participants who had joint/limb pain at baseline but only 6.8% of 148 participants who did not (*p* < 0.001). Furthermore, after 4 weeks, the severity of feeling cold had increased from baseline in 20% of the 20 participants who experienced more severe joint/limb pain compared to only 2.3% of the 171 in who did not (*p* < 0.001). However, it is not clear why the prevalence and severity of feeling cold would be less after 4 weeks on OBG vs. Control.

Hot flashes are a common symptom in post-menopausal women. In this study, 93.5% of the 46 participants with hot flashes at baseline were women (*p* < 0.001). Furthermore, although the mean ± SD age of women with and without hot flashes at baseline were similar, 52.5 ± 9.8 vs. 49.6 ± 11.4 year (*p* = 0.16), a higher percentage of *n* = 76 women aged > 48 year had hot flashes compared to the *n* = 43 aged ≤ 48 year, 46 vs. 19%, *p* = 0.003. Increased oxidative stress is associated with a greater severity of hot flashes [50], but it is not clear if menopausal symptoms are a cause or an effect of oxidative stress [51]. In this study the concentration of oxLDL at baseline (median {95%CI}) did not differ in the 46 participants with hot flashes at baseline, 63 {37, 124} μg/L, compared to the 145 without, 78 {45, 144} μg/L. However, it is of interest that the severity of hot flashes at week 4 compared to baseline increased more in participants in whom oxLDL increased compared to those in whom it did not (Figure 3D).

The major weakness of this assessment of non-GI symptoms is that the study was designed to assess the effect of OBG on serum cholesterol; assessment of symptoms was a tertiary objective not listed in the study registration. Several hundred statistical analyses were performed post hoc with no correction for multiple comparisons; this increases the likelihood of making type 1 errors. Furthermore, validated questionnaires were not used to assess the symptoms. Nevertheless, there was significant amount of internal and external consistency of the results with what might be expected from the literature.

## 5. Conclusions

These results provide hypothesis-generating evidence that OBG may have a beneficial effect on several affective and physical feeling states in healthy adults. Since there is a paucity of information about the effects of dietary fiber in general, and oats and OBG specifically, on non-GI symptoms in humans, these results provide information which may be useful for designing the studies which would be required to confirm these observations.

## Figures and Tables

**Figure 1 nutrients-13-01534-f001:**
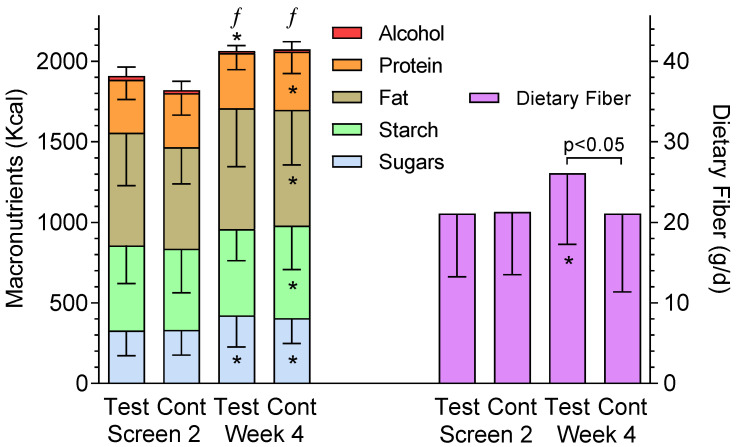
Macronutrient and dietary fiber intakes during the trial. Bars represent mean ± SEM intakes (expressed in Kcal for sugars, starch, fat, protein and alcohol) and grams for dietary fiber before treatment (dietary records obtained at the second screening visit; Screen 2) and during treatment (Week 4) for *n* = 96 on Test and *n* = 95 on Control (Cont). Error bars go down for sugars, starch, fat, protein and dietary fiber and up for alcohol. The only significant difference between Test and Control is for fiber. * Asterisks below the error bars indicate a significant difference from Screen 2 for the respective nutrient, except for alcohol the asterisk is above the error bar, (*p* < 0.05) by paired *t*-test. ƒ Significant difference in energy intake between Screen 2 and Week 4 (*p* < 0.05) by paired *t*-test.

**Figure 2 nutrients-13-01534-f002:**
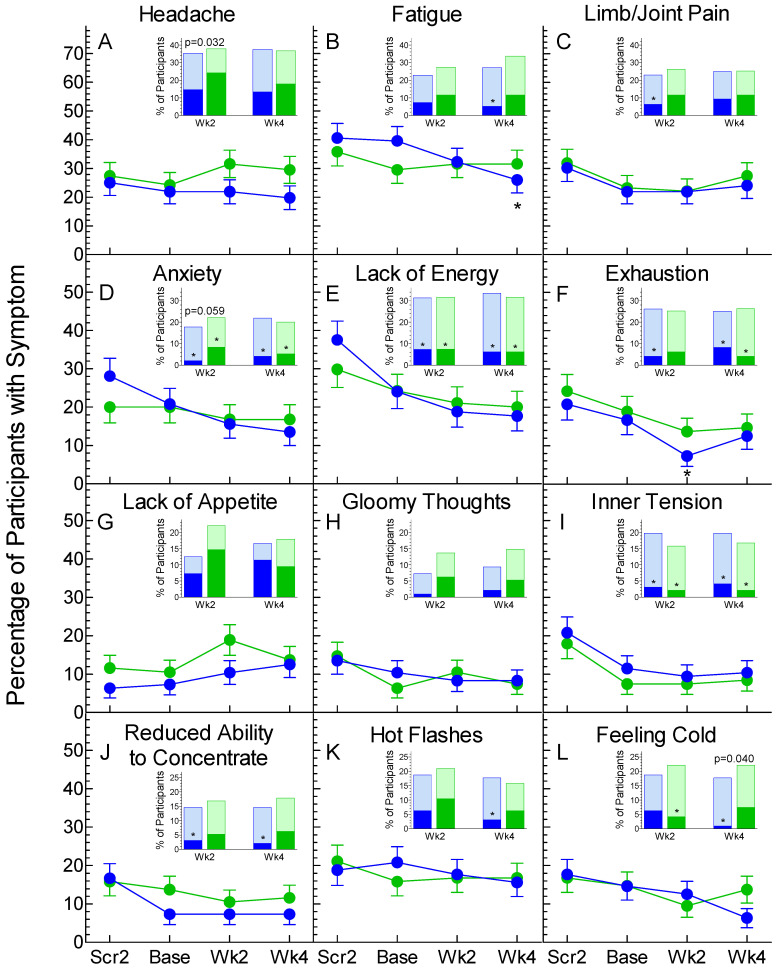
Prevalence and severity of headache (**A**), fatigue (**B**), limb/joint pain (**C**), anxiety (**D**), lack of energy (**E**), exhaustion (**F**), lack of appetite (**G**), gloomy thoughts (**H**), inner tension (**I**), reduced ability to concentrate (**J**), hot flashes (**K**) and feeling cold (**L**). Main panels: Values are means ± SD for *n* = 96 on Test (blue symbols and bars) and *n* = 95 on Control (green symbols and bars). Scr2 = second screening visit; Base = baseline; Week2 = week 2, Wk4 = week 4. * Prevalence significantly different from that at the Baseline visit by chi-squared test (*p* < 0.05). Insets: Percentage of participants in whom the symptom was more severe (dark bars) or less severe (light bars) than the severity before treatment (median severity at Scr2 and Base). The percent of participants with no change in severity is not shown. *p*-values are the significance of the difference between the ratio of M:L on Test vs. Control (*p* < 0.05 by Fisher’s exact test). * Ratio of M:L significantly different from 1:1 (*p* < 0.05 by binomial distribution) where M and L are the number of participants in whom the symptom was more severe (M) or less severe (L).

**Figure 3 nutrients-13-01534-f003:**
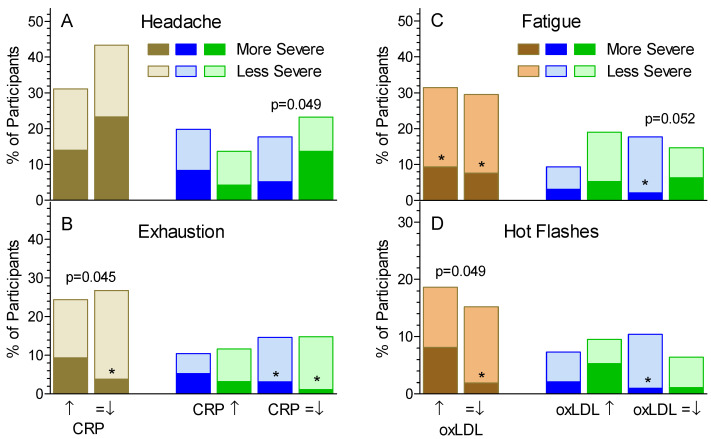
Effect of changes in serum CRP and oxLDL on symptom severity. Percent of participants whose symptoms became more severe (dark bars) or less severe (light bars) during treatment for those whose serum CRP (panels **A** and **B**) or oxLDL (panels **C** and **D**) increased (↑) or did not increase (=↓) after 4 weeks treatment. The percent of participants with no change in severity is not shown. Brown/orange bars are for all participants, blue bars for Test and green bars for Control. *p*-values over 2 bars are the significance of the difference between the ratios of More:Less severe for the 2 bars (e.g., panel A, in participants whose CRP did not increase after 4 weeks treatment, Headache was significantly more severe in those on Control vs. Test *p* = 0.049). * Ratio of more:less severe differs significantly from 1:1 by chi-square test (*p* < 0.05).

**Figure 4 nutrients-13-01534-f004:**
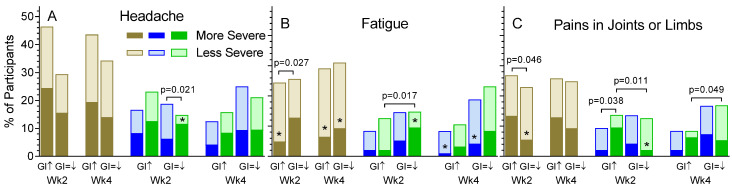
Effect of changes in GI symptoms on the severity of the non-GI symptoms headache (**A**), fatigue (**B**) and pains in joints or limbs (**C**). Bars show the percent of participants whose non-GI symptoms became more severe (dark bars) or less severe (light bars) during treatment for those whose GI symptoms (sum of scores for flatulence, diarrhea, constipation, abdominal distention, and abdominal pain) became more severe (GI↑) or not (GI = ↓) during treatment. The percent of participants with no change in severity is not shown. Brown bars show all participants, blue bars Test and green bars Control. *p*-values are the significance of the difference between the ratios of More:Less severe for the bars indicated (e.g., panel **A**, in participants whose GI symptoms did not become more severe after 2 weeks treatment, Headache was more severe on Control vs. Test *p* = 0.021). * Ratio of more:less severe differs significantly from 1:1 by chi-square test (*p* < 0.05).

**Table 1 nutrients-13-01534-t001:** Association of non-GI symptoms with major GI symptoms.

Non-GI Symptom	Major GI Symptom		*p*
	Absent (*n* = 103)	Present (*n* = 88)	
Headache	16 (16)	28 (32)	0.008
Fatigue	20 (19)	46 (52)	<0.001
Pains in joints or limbs	8 (8)	35 (40)	<0.001
Feelings of anxiety	12 (12)	27 (31)	0.001
Lack of energy	7 (7)	39 (44)	<0.001
Tend to become exhausted	8 (8)	26 (30)	<0.001
Lack of appetite	6 (6)	11 (13)	0.11
Gloomy thoughts	3 (3)	13 (15)	0.003
Inner tension	4 (4)	14 (16)	0.005
Diminished ability to concentrate	2 (2)	18 (20)	<0.001
Hot flashes	10 (10)	25 (28)	<0.001
Feeling cold	5 (5)	23 (26)	<0.001

Values are number (%) of participants with each non-GI symptom at baseline amongst the participants without any major GI symptom (flatulence, diarrhea, constipation, abdominal distension and abdominal pain) compared to those with one or more major GI symptom at screening or baseline. The right-most column shows the significance of the differences by chi-squared test.

## Data Availability

The data presented in this study are available upon request from the corresponding author.

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
