# Peer review of "Effect of Oat β-Glucan on Affective and Physical Feeling States in Healthy Adults: Evidence for Reduced Headache, Fatigue, Anxiety and Limb/Joint Pains"

_nutrients, 2021, doi:10.3390/nu13051534_

Round 1
Reviewer 1 Report
Review Effect on oat B-glucan on affective and physical feeling states in healthy adults: Evidence for reduced headache, fatigue, anxiety and limb/joint pains
The authors have produced a well written and an easy understandable manuscript. The subject is actual since many people tend to consume to low doses of dietary fibers, where oat is considered to have extra health effects due to its content of B-glucan. However, there are some major and minor concerns regarding the results and how the results are presented.
First of all; The authors refer to further details in the test protocol, Ref #9. However, that reference is submitted for publication elsewhere, and details cannot be further looked into by the reviewers. This is a major lack when reviewing the present manuscript, therefore missing information, referring to ref 9, must be included, or present manuscript must be put on hold until available.
Major concerns
- The approach to screen the participants before the actual measurement period is beneficial for reducing stress and increase comfort for the participants. Still it is very confusing in text, figure 1 and 2 to include both screen 2 and baseline as baseline values. It confuses the reader to read-out the actual start-value. In fig 1, week for of the test is compared to Sc2. In fig 2, both Scr2 and baseline is included. Note that the second screening visit is denoted differently in fig 1 and 2. In table S1 and S2, it is ok to present all data.
- The participants, which I assume, is a cross-section over the population in the area, includes, males, females, different ethnical backgrounds, different age, as well as a BMI spanning from 25.0 to <40. Since some of the non-GI outcome criteria are related to fatigue, exhaustion and limb/joint pain, it would have of major interest to look at those variables with a focus on BMI. This could be better outlined in table 1. Though it is partly considered, lines361-365.
- As described above, the paper would improve if participants would be compared, primarily within age and BMI. It might be that the readout from data would be more robust, and that differences regarding effects of B-glucan could be masked since all participants are included in test or control group.
- Since ref 9 is unavailable for the reviewers, the authors must describe how some of the symptoms were achieved. i.e how was fatigue and exhaustion measured or described?
- The authors should discuss the effect of B-glucan intake in the scope of the participants daily activity level, since it is well known that physical inactivity and GI health is closely related.
Minor
- Line 161. Inserted (Error! Bookmark not defined.)
- Figure 1. The bar should be associated at respective time point, and not by study group. This would increase comparison between the groups.
- Lines 223-228, These comparisons and significances are hard to follow in the text. Add it as a table to fig 2?
Author Response
The authors have produced a well written and an easy understandable manuscript. The subject is actual since many people tend to consume to low doses of dietary fibers, where oat is considered to have extra health effects due to its content of B-glucan. However, there are some major and minor concerns regarding the results and how the results are presented.
RESPONSE: thank you for your kind remarks.
First of all; The authors refer to further details in the test protocol, Ref #9. However, that reference is submitted for publication elsewhere, and details cannot be further looked into by the reviewers. This is a major lack when reviewing the present manuscript, therefore missing information, referring to ref 9, must be included, or present manuscript must be put on hold until available.
RESPONSE: we included all relevant information in the paper, but should have included a copy of the other unpublished manuscript (reference 9) with the submission so the reviewers could see the information referred to. The main manuscript (reference 9) has been reviewed and resubmitted after revision to address the minor corrections requested. A copy of that manuscript is included with this resubmission for the reviewers’ information.
Major concerns
- The approach to screen the participants before the actual measurement period is beneficial for reducing stress and increase comfort for the participants. Still it is very confusing in text, figure 1 and 2 to include both screen 2 and baseline as baseline values. It confuses the reader to read-out the actual start-value. In fig 1, week for of the test is compared to Sc2. In fig 2, both Scr2 and baseline is included. Note that the second screening visit is denoted differently in fig 1 and 2. In table S1 and S2, it is ok to present all data.
RESPONSE: Figure 1 shows the dietary intakes which were only assessed as Screen 2 and Week 4. We clarified the footnote to indicate that dietary intake before treatment was obtained from the diet records obtained at the Screen 2 and intake during treatment was obtained at Week 4. Based on your comment below, the graph was redrawn so the bars for Test and Control were next to each other for Screen 2 and Week 4.
Figure 2 shows the symptoms which were assessed at every visit (Scr2, Baseline, Wk2 and Wk4), and we feel it is useful to include all 4 assessments since, as the reviewer noted, simply paying attention to the participants may influence the presence of some symptoms and including all time points allows this to be assessed. The prevalence of symptoms at Wk2 and Wk4 is compared to that at Baseline; we clarified this in the footnote and in the Methods (lines 158-159). The severity of reported symptoms varied between Scr2 and Base, and we took the median of these 2 scores as the pre-treatment value with which to assess changes in severity during treatment, this is now more clearly explained in the methods (lines 161-165) and Results (lines 252-253) sections and in the Figure 2 footnote. We feel this is reasonable since all subjects consumed their usual diets and received the same amount and kind of attention at Scr2 and Base. An advantage of is that more subjects had a change in the severity of symptoms during treatment when compared to the median of the 2 scores before treatment than when compared the score at the baseline visit. This is because the symptom severity scores, (none, mild, moderate or severe) were assigned values of 0, 1, 2 or 3, respectively. Thus, if the scores at Scr2 and Base differed by an odd number (for example 0 and 1), the median is half a score (eg. here = 0.5), and since the scores at Wk2 and Wk4 are whole numbers, the median of the pre-treatment scores has to be difference from the scores at Wk2 and Wk4. With a higher n, there is more chance of being able to detect a significant difference (eg. 33% vs 67% is not significantly different by chi-squared test if each group only has n=12 (4:8 vs 8:4) but is significant if each group has n=18, 6:12 vs 12:6).
To further address this issue, we added to Supplementary Table S5 showing the severity of symptoms at Wk2 and Wk4 when compared to the score at the Baseline visit. As expected, the percentage of subjects with a change in the severity symptoms at Wk2 and Wk4 compared to Baseline is less than when compared to the median of Scr2 and Base. However, when compared to Baseline alone, the trends tend to be similar to those shown in Figure 2, but, due to the lower numbers, there are fewer significant differences.
2.The participants, which I assume, is a cross-section over the population in the area, includes, males, females, different ethnical backgrounds, different age, as well as a BMI spanning from 25.0 to <40. Since some of the non-GI outcome criteria are related to fatigue, exhaustion and limb/joint pain, it would have of major interest to look at those variables with a focus on BMI. This could be better outlined in table 1. Though it is partly considered, lines361-365.
RESPONSE: Thank you for this comment. We addressed it by adding Supplementary Table S4 showing the prevalence and severity of each symptom at Baseline by sex, age (<48 vs ≥48) and BMI (lean, <25; overweight, 25-<30; and obese >30), and, for each symptom, the mean age and BMI of participants with and without the symptom at Baseline. We also included in the Results section, a summary of the results (lines 240-246)
3. As described above, the paper would improve if participants would be compared, primarily within age and BMI. It might be that the readout from data would be more robust, and that differences regarding effects of B-glucan could be masked since all participants are included in test or control group.
RESPONSE: It is not clear what the reviewer means by saying: the readout from data would be more robust, and that differences regarding effects of B-glucan could be masked since all participants are included in test or control group. The distribution of sex, age and BMI were similar in the Test and Control groups and most of the symptoms were not influenced by age, sex or BMI, and this is now shown as requested above.
4. Since ref 9 is unavailable for the reviewers, the authors must describe how some of the symptoms were achieved. i.e how was fatigue and exhaustion measured or described?
RESPONSE: Reference 9 does not contain any information about what the non-GI symptoms were nor how they were measured. That information is described in section 2.3, lines 120-135. To this we added information about how the results for the 5 most prevalent GI symptoms calculated for presentation here. All symptoms were scored as either none, mild, moderate or severe.
5. The authors should discuss the effect of B-glucan intake in the scope of the participants daily activity level, since it is well known that physical inactivity and GI health is closely related.
RESPONSE: This is a good suggestion, but we did not collect any information about the amount of physical activity the participants did, so we cannot make any comments.
Minor
- Line 161. Inserted (Error! Bookmark not defined.)
RESPONSE: thank you for noticing this … it has been corrected.
2. Figure 1. The bar should be associated at respective time point, and not by study group. This would increase comparison between the groups.
RESPONSE: We think this means that the order of the bars from left to right should be Scr2 Test, Scr2 Control, Wk4 Test, Wk4 Control. Therefore, the Figure has been revised as suggested and the bars labeled “Test” and “Cont” at “Screen 2” and “Week 4”.
3. Lines 223-228, These comparisons and significances are hard to follow in the text. Add it as a table to fig 2?
RESPONSE: A table showing the number of individuals rating each symptom as none, mild, moderate or severe is shown in the Supplementary information as noted in lines 225-226.
Reviewer 2 Report
In this manuscript Wolever and colleagues evaluate the efficacy of oat beta glucan to alleviate an array of mental and physical symptoms vs placebo control in a cohort of participants with borderline-high LDL cholesterol. While the study seems to be well-conducted in some respects, there are crucial elements of the study design that need to be adequately explained before this manuscript can be seriously evaluated for publication, a control group should be added, and quality of some figures should be improved as outlined in the following major and minor comments:
Major:
- The Introduction lacks information regarding the following:
- Since the authors emphasize the LDL status of the study population, the pathway through which beta glucan reduces LDL should be explained, particularly as it relates to other fiber types.
- The authors mention fecal mass/bulk, but not why this is a beneficial or undesirable characteristic.
- The materials and methods need to include information or justification regarding the following points:
- What was used to record the 3-day diet record? Is this a validated questionnaire? Citation is required.
- It is understandable why participants consuming high amounts of fiber were excluded (since the intervention is to test the effects of fiber intake), but the exclusion of participants consuming high saturated fat needs to be justified. Isn’t saturated fat intake influential to LDL?
- A major weakness of this study is the lack of a control group that did not consume a supplement (neither “Test” nor “Control”) but simply participated by monitoring their psychological status during this time. It seems that self-monitoring and clinician attention/evaluation of symptoms, dietary intake, and the screening variables could lead to improvement, which is reflected in your outcomes of “Anxiety”, “Lack of Energy”, “Exhaustion”, and “Inner Tension”. At a minimum, references to studies that investigate such phenomena should be included, but inclusion of such a control group (even if temporally distinct) would drastically improve the study and claims made by the authors.
- In the discussion, the authors do a reasonable job of framing their results within the context of other literature, but do not propose specific pathways through which their intervention (beta glucans) altered physiological outcomes. For example:
- Why would the bacteria cited from other publications change gut-brain communication if not through SCFA?
- How do the bacteria contained in the probiotic formulation (line 378) compare to bacterial populations observed to change with beta glucan intake (beginning line 317)? This seems relevant to the highly microbiota-focused discussion.
- The discussion does not address the data from “Table 1”; are there previous reports that demonstrate physiological pathways linked between the observed mental and GI symptoms?
Minor:
- The inclusion of vague symptoms including “Gloomy thoughts” and “Inner tension” on the questionnaire raise doubt about its validity. Were these symptoms adequately described to the participants? Has this survey been properly validated by external psychologists or equivalent professionals? Clear explanation and citation is required.
- The text “(Error! Bookmark not )” appears in the “Statistical analysis” section.
- As currently presented, Figure 3 is difficult to understand.
- Is there a reason that a different color palette is used for A-B vs C-D?
- Shouldn’t the brown/orange bars sum to 100% (i.e. include all participants)? Should the green (or blue) bars sum to roughly 50%, or 100% (are they percent within group or within the study population)?
- Figure 4 has similar issues to Figure 3, with the addition of inconsistent labeling of increased/decreased GI symptoms (under brown bars).
- Was total LDL (or blood lipid panel) not measured in these samples? This seems highly relevant to the results, particularly since the authors explain that differences in SCFA are unlikely to be mediating these responses.
- Why were the results of this study not published in the same manuscript as the GI symptom-focused manuscript if one of the objectives was to relate non-GI symptoms to GI symptoms, particularly if all of the data was collected in tandem?
- The use of the term “significant” throughout should be assumed based on the statistical methods and is therefore redundant.
Round 2
Reviewer 1 Report
The authors have thoroughly revised the manuscript according to reviewers comments, and can be accepted as it is.
Reviewer 2 Report
Thank you for your resubmission and responses.
I can see that the authors have made some effort to improve the manuscript based upon my comments.
I agree that their placebo control is an ideal element of the study design. However, I still think that a "medical observation" control group would vastly strengthen the claims made in the paper (and future studies), particularly since their placebo group experienced improvements in multiple non-GI symptoms over the course of the study, and the authors make many references to the gut-brain axis. Were improvements due to the placebo supplement itself, or the attention given to the subjects? This is important to distinguish between the positive effects coming from OBG-consumption, vs consuming any soluble carbohydrate powder (placebo effect?), vs attention given to subjects - which is demonstrated to improve cognitive symptoms (see https://doi.org/10.1186/s12916-017-0791-y as an example). However, for the emphasized "preliminary" nature of this manuscript, the extra control group is not strictly necessary for publication.
While I do not agree, it is not my job as a reviewer to force my views on the authors, only make suggestions that would improve the article and ensure that the methodology and rationale are substantiated - which the authors have done here. Ultimately it is the decision of the editors to decide what content is published in their journal.